# Development of Strategies to Minimize the Risk of *Listeria monocytogenes* Contamination in Radish, Oriental Melon, and Carrots

**DOI:** 10.3390/foods10092135

**Published:** 2021-09-09

**Authors:** Nagendran Rajalingam, Hyo-Bin Chae, Hyeon-Jin Chu, Se-Ri Kim, Injun Hwang, Jeong-Eun Hyun, Song-Yi Choi

**Affiliations:** Microbial Safety Team, National Institute of Agricultural Sciences, Rural Development Administration, Wanju 55365, Korea; nagendranrajalingam@gmail.com (N.R.); hyobin3811@korea.kr (H.-B.C.); chj07079@korea.kr (H.-J.C.); seri81@korea.kr (S.-R.K.); injun0370@korea.kr (I.H.); jeongeunh@korea.kr (J.-E.H.)

**Keywords:** *Listeria monocytogenes*, postharvest washing, microbial contamination, agricultural products, sanitizers, organic acids, ethanol, UV-C

## Abstract

Contamination by *Listeria monocytogenes* in packaged produce is a major concern. The purpose of this study was to find natural and affordable sanitizers to reduce *L. monocytogenes* contamination in agricultural products. Organic acids, ultraviolet-C (UV-C), and ethanol were analyzed either alone or in combination to assess their ability to reduce *L. monocytogenes* population in radish, oriental melon, and carrot samples. In radish samples, 3% malic acid combined with UV-C at a dosage of 144 mj/cm^2^ significantly reduced (>4 log CFU/g) the population of *L. monocytogenes* (1.44 ± 0.5) compared to the control sample (5.14 ± 0.09). In the case of the melon samples, exposure to UV-C at a dosage of 144 mj/cm^2^ combined with 3% lactic acid (2.73 ± 0.75) or 50% ethanol (2.30 ± 0.01) was effective against *L. monocytogenes* compared to the control sample (5.10 ± 0.19). In carrot samples, 3% lactic acid combined with 144 mj/cm^2^ dosage UV-C reduced *L. monocytogenes* population (4.48 ± 0.25) more than in the control sample (5.85 ± 0.08). These results reveal that sanitizers that are effective for one crop are less effective for another crop indicating that effective prevention methods should be customized for each crop to prevent pathogen cross contamination during postharvest washing.

## 1. Introduction

*Listeria monocytogenes,* a psychotropic pathogen associated with consumption of fresh produce is one of the most important foodborne pathogens that causes listeriosis, a serious life-threatening invasive disease [1]. According to the Centers for Disease Control and Prevention (CDC), an estimated 1600 people are diagnosed with listeriosis and about 260 people die every year [2]. Immunocompromised individuals, elderly people over 65, and pregnant women and their neonates are most likely to be affected by listeriosis [3]. The number of agricultural produce related *L. monocytogenes* outbreaks has increased in recent years. For instance, the United States Food and Drug Administration (FDA), along with the CDC, reported several multistate outbreaks of *L. monocytogenes* infections linked to enoki mushrooms [4], prepackaged caramel apples [5], and frozen vegetables [6] in the past five years. Melon-associated *L. monocytogenes* outbreaks have also been reported in previous years [7,8]. In addition, the prevalence of *L. monocytogenes* in fresh produce including lettuce [9], mushrooms [10], cabbage [11], carrots [12,13], radishes [14,15], corn [16], parsley [17], strawberries [10], watercress [18], celery [19], red peppers [20], cucumbers [21], and several other salad vegetables [22] has been reported through microbiological risk assessment surveys from various countries.

Agricultural products can acquire pathogens through several sources during farming (contaminated soil, raw materials, livestock manure, irrigation water, pets, or farm workers) or at any step during the postharvest process (peeling, trimming, cutting, washing, and packing) [23]. The postharvest washing step at a commercial scale removes field-acquired contamination such as dirt, chemicals, cell exudates, and reduces microbial load up to 1- to 2- log [24]. However, most pathogens attach to the food surface firmly and form biofilms [25] or become internalized in the edible parts of the plant that are hard to remove by rinsing with cold water [26]. In some cases, the washing water can be a source of cross contamination that can lead to the distribution of pathogens between clean and contaminated food products [27]. Hence, washing the food products with tap water alone in a water bath is inefficient to control pathogen contamination unless effective sanitizers are combined with water [28].

Developing prevention technologies to eradicate microbial contamination in agricultural products is very crucial. Several chemical (chlorine [29], chlorine dioxide [30], paracetic acid [31], ozone [32,33], sodium hypochlorite, and acidified sodium chlorite [34]) methods have been applied on food products to control *L. monocytogenes* contamination. However, these methods can lead to disinfection byproduct (DBPs) formation, and chemical residues left on the food surfaces after washing could affect the quality and safety of the food products and may cause serious health risks to consumers [35]. Nonthermal food processing physical methods such as high hydrostatic pressure [36], pulsed light [37], ultrasound [38,39], plasma-activated water [40], cold plasma-activated hydrogen peroxide aerosol [41], dielectric barrier discharge gas plasma [42,43], low dose irradiation [44], etc., have been shown to be efficient in inactivating *L. monocytogenes* and other foodborne pathogens in fresh produce. However, these methods may require huge investments [45] and high operational costs [46] along with more processing time [47], which may be reflected the cost of the food products. Hence, it is necessary to find natural, affordable, and nontoxic interventions that can eliminate pathogens. Nevertheless, any disinfection method can only reduce the pathogen population in contaminated fresh produce but cannot completely decontaminate produce.

Organic acids can be used as alternatives for chlorine based chemical sanitizers as they are generally recognized as safe (GRAS) to add to foods [28]. Antimicrobial properties of organic acids depend on their dissociation constants or pKa values [48]. At low pH, the undissociated portion of the organic acids diffuse through the cell membranes and alter the internal chemical equilibrium of the microbial cells, which severely impairs their metabolism [49] and other cellular processes such as DNA replication and protein synthesis leading to cell death [50]. Ultraviolet (UV-C) treatment is one of the most effective method to inactivate pathogens as it is nonthermal and does not leave any toxic residues in foods [51] and consumer safe as it does not alter the properties of the food products [52]. UV-C radiation impairs the genomic DNA of bacteria by inducing pyrimidine dimers, which consequently inhibits transcription and replication, which eventually causes cell death [53]. Ethyl alcohol (ethanol) is broadly used in many research and clinical labs as disinfectants to kill microbes. In addition, ethanol produced during fermentation of foods and drinks has been shown to have a preservative function against microorganisms [54]. Ethanol increases membrane permeability and sensitizes microbial cells to osmotic stress and pH stress leading to cell death [55].

The efficiency of sanitizers depend on the type of food and characteristics of the sanitizing method applied [56]. Although the standalone bactericidal effects of organic acids, UV-C, and ethanol have been well documented on fresh produce [48,57,58], studies regarding their combinational effects and application during produce washing is still limited. In this study, we analyzed the efficiency of organic acids, UV-C, and ethanol either alone or in combination to reduce *L. monocytogenes* population in radish, oriental melon, (melon) and carrot samples to minimize the risk of disease outbreak through agricultural products. As all the disinfection strategies were performed on a laboratory scale without pilot scale trials, several physiochemical parameters in these methods should be optimized in detail before implementing in an industrial setup.

## 2. Materials and Methods

### 2.1. Sample Preparation and Inoculation

Three different isolates of *Listeria monocytogenes* (NX1-serotype 3c str. 10-5027 [Genome ID: CP007196]; 2YC-serotype 1/2a str. 10-0934 [CP007200]; B2-FORC_049 [CP016629]) were revived from glycerol stock (−80 °C) on tryptic soy agar–yeast extract (TSA–Ye; Oxoid Ltd., Basingstoke, UK) plates prior to each experiment. Single colonies of the bacteria from the TSA–Ye plates were picked and inoculated separately in tubes containing tryptic soy broth–yeast extract (TSB–Ye; Oxoid Ltd., Basingstoke, UK) and incubated at 37⁰C. After 24 h, the bacterial cells were collected by centrifugation at 3000× *g* for 10 min (Labogene 1736R, Gyrozen Co. Ltd., Seoul, Korea) and washed 2 times with 1 mL phosphate-buffered saline (PBS; Biosesang, Seongnam, Korea) (pH 7.3) buffer suspension. The washed isolates were mixed together as a cocktail after centrifugation by suspending the pellet with 5 mL PBS buffer up to a concentration of 0.5 OD_600_ (1.5 × 10^7^ CFU/mL).

The radish, melon, and carrot samples purchased from local markets in Korea were warmed to room temperature and cut into small pieces [carrot-2 × 2 cm (8–10 g); melon-2 × 5 × 1 cm (10–14g); radish-2 × 2 cm (8–10 g)] (Figure 1A). The cut vegetables were treated with 70% ethanol for 5 min, followed by washing with sterile distilled water [59], and dried on a clean bench until the vegetables were completely dried (30 min–1 h). The cut vegetables were spot-inoculated with 100 μL of *L. monocytogenes* cocktail suspension (10 μL in 10 spots around each piece of vegetable). After spot-inoculation, the cut vegetables were dried on a clean bench at room temperature for around 30 min to 1 h/until it is dried. For experiments with whole (intact) vegetables (Figure 1B), bacterial suspensions were sprayed on the vegetable surfaces and air-dried before treatment. Equal volumes of spray [59] collected in 15 mL falcon tubes were used as control.

### 2.2. Individual Treatments

Organic acids [Acetic acid (pH-2.38), L(+)-Ascorbic acid (pH-2.36), Citric acid anhydrous (pH-1.80), and DL-Malic acid (pH-2.38) (Daejung Chemicals and Metals Co. Ltd., Siheung, Korea)], Lactic acid (pH-2.40) (Kanto Chemical Co. Inc., Tokyo, Japan), and L(+)Tartaric acid (pH-1.82) (Sigma–Aldrich Inc., Merck KGaA, Darmstadt, Germany) prepared at 1% and 3% concentrations were filter sterilized with 0.22 µm pore size MF-Millipore™ membrane filters (Merck KGaA, Darmstadt, Germany). To investigate the individual efficacy of organic acids against *L. monocytogenes* in carrot, melon, and radish samples, the *L. monocytogenes* spot-inoculated cut vegetables were immersed (dipping) in a sterile filter bag containing 60 mL of the organic acids for 5 min. For ethanol treatment, the cut vegetables were dipped in sterile filter bags containing 60 mL of food-grade fermented ethanol (Ethanol Supplies World Co., Ltd., Jeonju, Korea) prepared at different concentrations (5%, 10%, 20%, 50%, and 70%) for 1 and 3 min. The efficacy of UV-C (254 nm) treatment was analyzed on cut vegetables with different intensities [4.7 mW/cm^2^ (15 cm distance), 2.4 mW/cm^2^ (30 cm), and 1.2 mW/cm^2^ (45 cm)] and time points (5, 10, 15, and 30 min) (Figure 1A). For control treatment, the cut vegetables were treated with 60 mL DW in sterile filter bags for 5 min. After treatment, the cut vegetables were immediately transferred to a stomacher bag containing 9x volume of BD Difco™ D/E (Dey-Engley) neutralizing broth (D/E broth; Difco, Becton Dickinson Co., Sparks, MD, USA) to neutralize the residual effects of sanitizers. All the treatments were performed at room temperature (25 °C).

### 2.3. Combined Treatments

Based on the results of individual treatments, we analyzed the combined efficacy of the following treatments: organic acids, malic acid, or lactic acid at 3% concentrations combined with UV-C at 30 cm distance for 1 min (144 mj/cm^2^); a combination of 1.5% malic acid or 1.5% lactic acid with or without UV-C (144 mj/cm^2^), and 50% ethanol combined with UV-C (144 mj/cm^2^). The combined treatments were tested on cut vegetables (dipping and spraying) and intact vegetables (spraying) (Figure 1B). After treatment, the cut vegetables or the peels (~10 g) from intact vegetables separated with a sterile peeler were immediately transferred to a stomacher bag containing 9× volume of BD Difco™ D/E (Dey-Engley) neutralizing broth and homogenized with a BagMixer^®^ 400 (Interscience, St. Nom, France) stomacher for 1 min.

### 2.4. Bacterial Enumeration

The homogenized filtered mixture was serial diluted with 9 mL of 0.1% peptone water (PW; Oxoid). One hundred microliters of the serial diluted samples were spread on PALCAM agar base (Oxoid) plates and incubated at 37 °C. After incubation for 24 h, the bacterial colonies were enumerated.

### 2.5. Statistical Analysis

All the treatments were repeated three times (biological replicates) with triplicates (technical replicates). Each biological replicate involving inoculated radish, melon, and carrot samples consisted of three samples (cut vegetable for individual treatments and intact vegetable for combination treatments) exposed to the same treatment conditions. Means and standard deviations were evaluated using the Microsoft Excel software (Microsoft Corporation, Redmond, WA, USA). Bacterial enumeration (CFU/g of sample) were analyzed after log transformation. The SigmaPlot software version 12.5 (Systat Software Inc., Chicago, IL, USA) was used to perform t-tests to assess the significant variation between the treated cut/intact vegetable samples and untreated control samples. Statistics by *t*-test are represented as * *p* < 0.01, ** *p* < 0.001, and *** *p* < 0.0001 [60].

## 3. Results

### 3.1. Efficiency of Organic Acids against Listeria monocytogenes in Fresh Produce

The populations of *L. monocytogenes* in untreated control radish, melon, and carrot cut samples were 7.69 ± 0.12, 6.18 ± 0.02, and 6.22 ± 0.02 log CFU/g, respectively. Malic acid and lactic acid of 3% were most effective against *L. monocytogenes* in radish (5.53 ± 0.18 and 4.53 ± 0.04 log CFU/g) and melon (2.37 ± 0.07 and 1.19 ± 0.39 log CFU/g) samples compared to the untreated radish and melon control samples. The reduction in *L. monocytogenes* by 3% malic acid and 3% lactic acid in radish samples were >2 log CFU/g and in melons were >4 log CFU/g (Figure 2). In the case of the carrot samples, a 1 log CFU/g reduction (5.15 ± 0.05 log CFU/g) in the population of *L. monocytogenes* was observed between the control and treated samples.

### 3.2. Efficiency of UV-C against Listeria monocytogenes in Fresh Produce

UV-C at intensities of 4.7 mW/cm^2^ and 2.4 mW/cm^2^ were effective (>2 log CFU/g reduction) in radish samples at 1 min (5.72 ± 0.23 log CFU/g) and 5 min (5.77 ± 0.11 log CFU/g), respectively. In melon samples, UV-C radiation of 2.4 mW/cm^2^ for 15 min (4.85 ± 0.21 log CFU/g) and 30 min (3.03 ± 0.48 log CFU/g) exposure were the most effective (>2 and 4 log CFU/g reduction) in reducing *L. monocytogenes*. In the case of the carrot samples, exposure of 4.7 mW/cm^2^ UV-C for 1 min (6.61 ± 0.22 log CFU/g) showed a 1 log CFU/g reduction in *L. monocytogenes*. The levels of *L. monocytogenes* in the untreated control radish, melon, and carrot samples were 7.82 ± 0.12, 7.48 ± 0.10, and 7.71 ± 0.07 log CFU/g, respectively (Figure 3).

### 3.3. Efficiency of Ethanol against Listeria monocytogenes in Fresh Produce

The populations of *L. monocytogenes* in untreated control radish, melon, and carrot samples were 8.10 ± 0.09, 6.63 ± 0.06, and 7.10 ± 0.06 log CFU/g, respectively. Ethanol at concentrations of 50% and 70% for 1 min (4.23 ± 0.14 and 4.10 ± 0.83 log CFU/g) and 3 min (4.20 ± 0.06 and 2.97 ± 0.25 log CFU/g) were most effective (>3.8 log CFU/g reduction) in radish samples. In the case of the melon samples, 50% ethanol for 3 min (4.21 ± 0.28 log CFU/g) and 70% ethanol for 1 min (4.06 ± 0.04 log CFU/g) and 3 min (3.60 ± 0.89 log CFU/g) were effective (>2.5 log CFU/g reduction). *L. monocytogenes* was significantly reduced (1.3 log CFU/g reduction) in carrots when 50% ethanol was applied for 3 min (5.80 ± 0.47 log CFU/g) (Figure 4).

### 3.4. Combination Efficiency of Organic Acids, UV, and Ethanol against Listeria monocytogenes on Cut and Intact Vegetables

The organic acids, malic acid or lactic acid, at 3% concentrations combined with UV-C at 30 cm distance for 1 min (144 mj/cm^2^); a combination of 1.5% malic acid or 1.5% lactic acid with or without UV-C (144 mj/cm^2^), and 50% ethanol combined with UV-C (144 mj/cm^2^) were tested for their efficiency against *L. monocytogenes* in cut vegetable samples of radish, melon, and carrot. The treatments were applied to cut vegetables by the dipping method. The densities of *L. monocytogenes* inoculated cut vegetables without dipping treatment in control radish, melon, and carrot samples were 7.79 ± 0.04, 7.67 ± 0.07, and 7.36 ± 0.05 log CFU/g, respectively. Combinations of 1.5% malic acid + 1.5% lactic acid + UV-C 30 cm/1 min (144 mj/cm^2^) showed >4 log CFU/g reduction in radish, melon, and carrot samples. Malic acid (3%) + UV-C (144 mj/cm^2^) reduced >5 log CFU/g in radish and melon samples while >4 log CFU/g reduction was observed in carrot samples. Lactic acid (3%) + UV-C (144 mj/cm^2^) significantly reduced *L. monocytogenes* in radish (>4.94 log CFU/g), melon (>5.67 log CFU/g), and carrot (>2.4 log CFU/g) samples. Ethanol (50%) + UV-C (144 mj/cm^2^) treatment by dipping method effectively reduced *L. monocytogenes* by >2.07 log CFU/g in radish, >4.67 log CFU/g in melon, and >4.34 log CFU/g in carrot samples (Figure 5).

The dipping treatment of cut vegetables in either 3% malic acid, 3% lactic acid, or 50% ethanol along with UV-C (144 mj/cm^2^) was highly efficient against *L. monocytogenes*. However, applying the dipping method to cut vegetables on a commercial scale is impracticable. Hence, we employed a spraying method on intact vegetables to analyze the efficiency of combined treatments (Figure 1). In radish samples, 3% malic acid + UV-C (144 mj/cm^2^) significantly reduced (>4 log CFU/g) the population of *L. monocytogenes* (1.44 ± 0.5) compared to the control sample (5.14 ± 0.09 log CFU/g) (Figure 6). In the case of melon samples, 3% lactic acid + 144 mj/cm^2^ UV-C (2.30 ± 0.75 log CFU/g) and 50% ethanol + 144 mj/cm^2^ UV-C (2.30 ± 0.01 log CFU/g) were effective against *L. monocytogenes.* Lactic acid (3%) + UV-C (144 mj/cm^2^) showed a slight reduction (4.48 ± 0.25 log CFU/g) in carrot samples.

## 4. Discussion

The attachment, growth, and survival of *L. monocytogenes* on the surfaces of vegetables vary between crops based on their surface characteristics [61], storage time, and temperature where the crop is produced and processed [19]. The antimicrobial effects of organic acids, UV-C, and ethanol on lettuce, apples, and other vegetables have been documented previously [3,28,57,58]. However, studies regarding the individual and combinational efficiency of these treatments on other vegetables during produce washing is still limited. This is the first study to investigate the individual and combined efficiency of organic acids, UV-C, and ethanol to reduce *L. monocytogenes* contamination on carrot, radish, and melon.

Organic acids are considered as safe and potential alternatives for chlorine based chemical sanitizers owing to their preservative and antioxidant properties and most importantly, their natural origin as well as low production cost [28]. Moreover, the disinfection efficiency of organic acids were proved to be on par or better than some chlorine based chemical sanitizers [62]. In our study, among all the other organic acids (acetic acid, ascorbic acid, citric acid, and tartaric acid), malic acid and lactic acid at a concentration of 3% were more effective in reducing *L. monocytogenes* population in radish (2 and 3 log CFU/g reduction) and melon (4 and 5 log CFU/g reduction) samples (Figure 2). In a similar study on enoki mushrooms, Kim et al. (2020) observed 3 log reductions in *L. monocytogenes* when treated with 3% lactic acid and malic acid for 10 min [63]. Concha-Meyer et al. (2017) observed a complete inhibition of *L. monocytogenes* in solutions of tryptic soy broth with yeast extract (TSB–YE) containing malic acid and lactic acid [64].

This disinfection ability of malic acid and lactic acid could be due to their smaller undissociated molecular sizes (134.09 Da and 90.08 Da, respectively), which may easily diffuse through the microbial cell membrane and change their internal pH [65]. This change in pH causes deformation and obstruction to intracellular activities and eventually cell death [55]. Their counterparts ascorbic (176.12 Da), citric (192.13 Da), and tartaric (150.09 Da) acids did not effectively reduce *L. monocytogenes* possibly due to their larger undissociated molecular sizes [66]. As a case in point, transmission electron microscopy studies by Raybaudi-Massilia et al. (2009) and Yoon et al. (2021) revealed that malic acid caused damage in the cytoplasm of *L. monocytogenes*, *S. enteritidis* and *E. coli* without altering the structure of their cell membranes [67,68]. Though the undissociated molecular size of acetic acid (60 Da) is smaller than the malic acid and lactic acid, its inefficiency in killing *L. monocytogenes* is yet unknown. In addition to the smaller molecular size, the acidity of malic acid was shown to be less affected by the organic substrates released from the crops during treatments making them more effective in reducing *L. monocytogenes* than other organic acids [68].

The bactericidal efficiency of UV-C light on fruit surfaces differs based on the surface characteristics. Fruit surfaces with rough texture help pathogens to attach firmly [69,70] to the fruits making them difficult to access and remove during sanitizing treatments [71,72]. Moreover, UV-C light cannot penetrate through the bumps and cracks on rough surfaces. Silva et al. (2003) demonstrated that UV light was not able to probe through the cracks and protuberance of low-density polyethylene film and effectively reduce *Staphylococcus aureus* and *E. coli* populations on them [73]. Adhikari et al. (2015) observed significant reduction in *L. monocytogenes* population on apple and pear fruits that have smoother surfaces compared to cantaloupe and strawberry that have rough surfaces by UV-C light exposure [74]. Correspondingly, in our study, UV-C treatment significantly reduced *L. monocytogenes* on radish (2-log CFU/g reduction at a dosage of 144 mj/cm^2^) and melon (3-log CFU/g reduction at a dosage of 144 mj/cm^2^) samples (Figure 3), which have smooth surfaces while no significant reduction was observed on carrot surfaces with coarse texture. Since UV-C light may not be able to access through the surfaces of carrots efficiently as that of radish and melon samples, their effectiveness in inactivating *L. monocytogenes* was different between the samples.

Ethanol at a concentration of 50% and 70% effectively inactivated 2 to 4 log CFU/g of *L. monocytogenes* in radish and melon samples in 3 min (Figure 4). Ethanol has been frequently used as a sanitizer in several clinical and microbiological labs for several years. However, its individual application on food products as a disinfectant is exceedingly rare. In many cases, ethanol was mostly combined with other sanitizers to use on food products. For instance, ethanol at a concentration of 30% combined with 1% ascorbic acid effectively inactivated pathogens on fresh-cut apples [75] and fresh-cut lotus root [76]. In another study, 5% ethanol with low pH (3.0) combined with 50 mM formate significantly reduced *L. monocytogenes* in 4 min [55]. Apart from inactivating the microbes, ethanol evaporates quickly without leaving any residue on the fruit surface making it a suitable candidate for sanitizing treatments.

UV-C treatment has limited efficiency in inactivating microbes on rough surfaces and the application of organic acids at low concentrations would be more preferable than higher concentrations. Based on the practical factors such as minimal processing time and low concentration, we speculated that combination of these treatments could increase the efficiency of sanitizers against *L. monocytogenes* on food products. Dipping of cut vegetables in either malic acid (1.5%) + lactic acid (1.5%), 3% malic acid, 3% lactic acid or 50% ethanol along with UV-C (144 mj/cm^2^) significantly reduced *L. monocytogenes* in radish, melon, and carrot samples (Figure 5). However, applying the dipping method to cut or intact vegetables is not feasible in a commercial scale because cross contamination may occur between contaminated and noncontaminated fresh produce and a change in the pH of dipping solution due to release of naturally occurring organic acids from vegetables may hinder disinfection efficiency [77]. Hence, we employed a spraying method for analyzing the efficiency of combined treatments on cut and intact vegetables. Since no significant reduction in *L. monocytogenes* was observed on cut vegetables by the spraying method (data not shown), we tested the combined efficiency of the treatments on intact vegetables. Combined treatments of 3% malic acid with UV-C at a dosage of 144 mj/cm^2^ significantly reduced (>4 log CFU/g reduction) *L. monocytogenes* in radish samples (Figure 6). Lactic acid at a concentration of 3% combined with UV-C (144 mj/cm^2^ dose) or 50% ethanol with UV-C (144 mj/cm^2^) significantly reduced *L. monocytogenes* in melon (>2 log CFU/g reduction) samples. Interestingly, a combination of 3% lactic acid with UV-C (144 mj/cm^2^) reduced *L. monocytogenes* up to 1.5 log CFU/g in carrot samples where individual application of these treatments did not significantly inactivate *L. monocytogenes.*

According to the results of our experiment, spraying 3% lactic acid or 50% ethanol during washing followed by UV-C (144 mj/cm^2^) application can reduce the risk of *L. monocytogenes.* However, there are some limitations in this study to using these strategies on an industrial scale. For instance, the organic acids lactic acid or malic acid can assist in reducing pathogenic contamination but cannot completely inactivate or kill the pathogens in contaminated fresh produce [48,77]. Although the UV-C and ethanol treatments were effective in reducing *L. monocytogenes* in this study, application of these methods in an industrial setting is not common due to complex management, additional cost, technical issues, and other practical factors [78]. In addition, all the experiments in this study were performed in a laboratory setting at room temperature while most of the fresh-cut processing in industries is carried out at temperatures below 10 °C.

Taken together, the results of this study reveal that sanitizing treatments developed against microbes for one food product are often less effective for another food product. This could be due to the difference in the characteristics of the food surfaces, which plays a vital role in the effectiveness of the sanitizers [77]. Several other attributes including exposure time, contact angle, surface topology, hydrophobicity, morphology, and cell wall components of the fruit surfaces also influence the antimicrobial efficiency of the sanitizers [73,79]. Hence, a better understanding of the surface characteristics of each agricultural product can help to develop an efficient customized sanitizing treatment to prevent pathogen contamination during the washing step in the production process.

## Figures and Tables

**Figure 1 foods-10-02135-f001:**
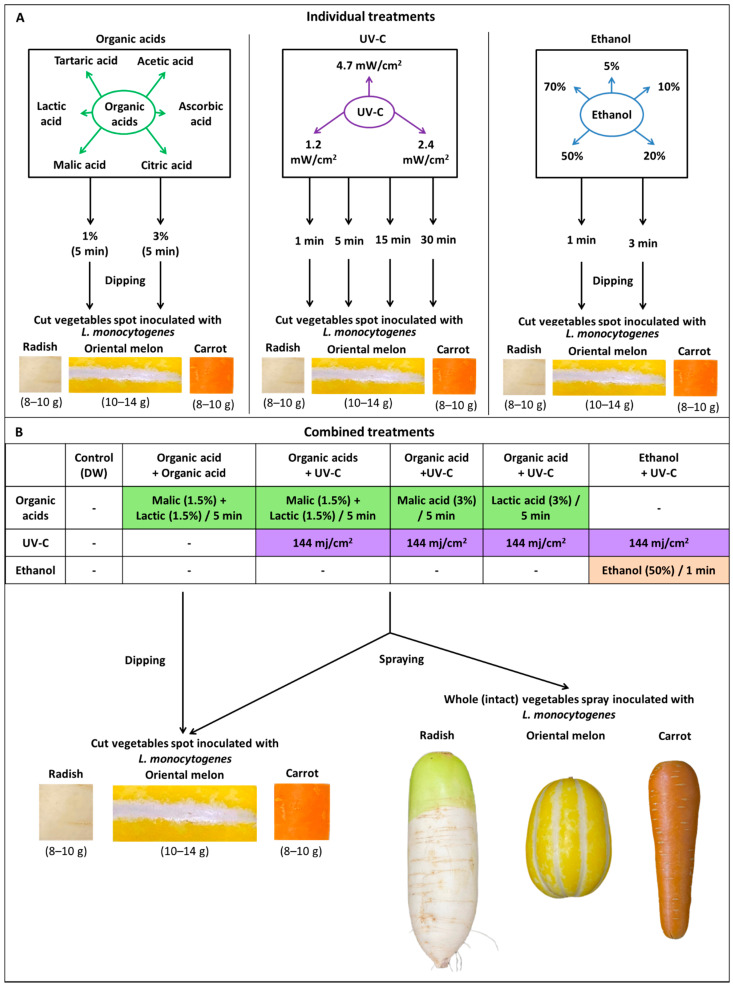
Schematic representation of the different treatment strategies that were applied on cut and intact vegetables. The individual (**A**) and combined (**B**) efficacy of organic acids, UV-C, and ethanol against *L. monocytogenes* were analyzed on cut and intact radish, oriental melon, and carrot samples.

**Figure 2 foods-10-02135-f002:**
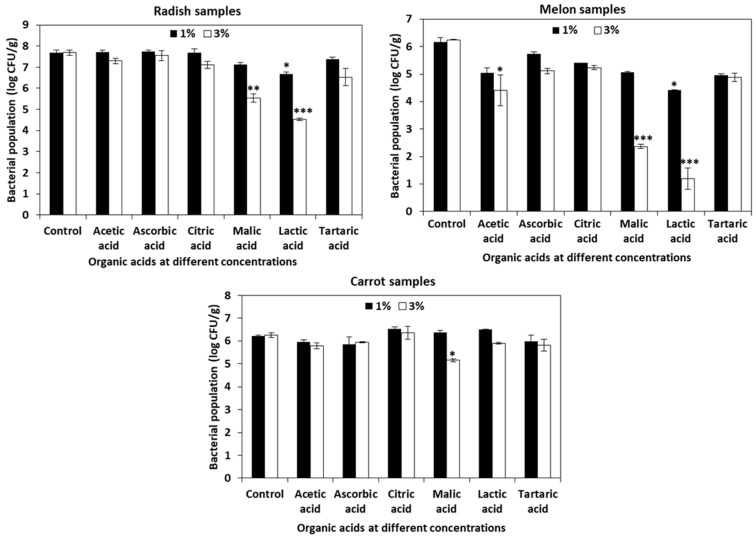
Efficiency of organic acids against *L. monocytogenes* in cut vegetable samples of radish, melon, and carrot. The organic acids acetic acid, ascorbic acid, citric acid, malic acid, lactic acid, and tartaric acid at a concentration of 1% and 3% were applied on radish, melon, and carrot samples and their effectiveness against *L. monocytogenes* was analyzed. The data obtained are the means ± SD from three independent experiments with triplicates. Statistical analysis (*t*-test) was conducted using SigmaPlot. For t-test analysis, the values of organic acids-treated cut vegetable samples were compared with those of the untreated cut vegetable control samples. An asterisk indicates significant differences compared with the control (* *p* < 0.01, ** *p* < 0.001, *** *p* < 0.001).

**Figure 3 foods-10-02135-f003:**
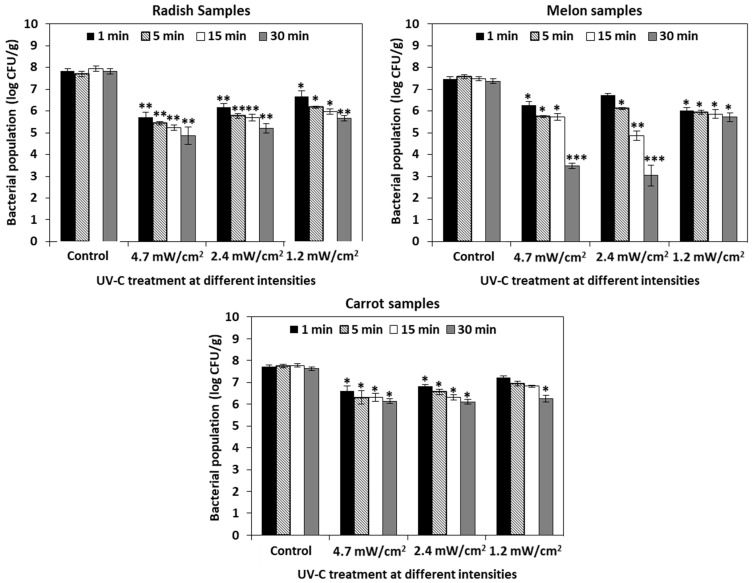
Effectiveness of UV-C against *Listeria monocytogenes* in cut vegetable samples of radish, melon, and carrot. Radish, melon, and carrot samples were exposed to UV-C radiation at a distance of 15, 30, and 45 cm for 1, 5, 15, and 30 min and the population of *L. monocytogenes* was enumerated on PALCAM agar plates. The data obtained are the means ± SD from three independent experiments with triplicates. Statistical analysis (t-test) was conducted using SigmaPlot. For t-test analysis, the values of the UV-C-treated cut vegetable samples were compared with that of the untreated cut vegetable control samples. An asterisk indicates significant differences compared with the control samples (* *p* < 0.01, ** *p* < 0.001, *** *p* < 0.001).

**Figure 4 foods-10-02135-f004:**
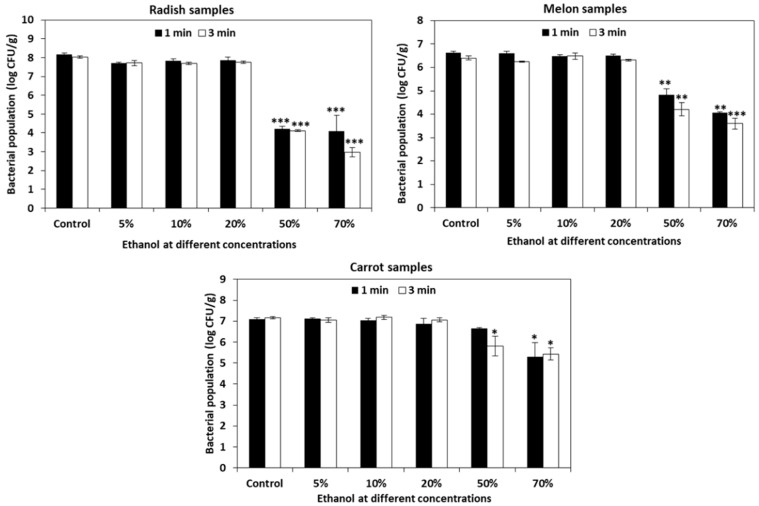
Efficiency of ethanol against *Listeria monocytogenes* in cut vegetable samples of radish, melon, and carrot. Ethanol at different concentrations (5%, 10%, 20%, 50%, and 70%) was applied on radish, melon, and carrot samples for 1 and 3 min and its effectiveness against *L. monocytogenes* was analyzed. The data obtained are the means ± SD from three independent experiments with triplicates. Statistical analysis (t-test) was conducted using SigmaPlot. For *t*-test analysis, the values of ethanol-treated cut vegetable samples were compared with those of the untreated cut vegetable control samples. An asterisk indicates significant differences compared with the control (* *p* < 0.01, ** *p* < 0.001, *** *p* < 0.001).

**Figure 5 foods-10-02135-f005:**
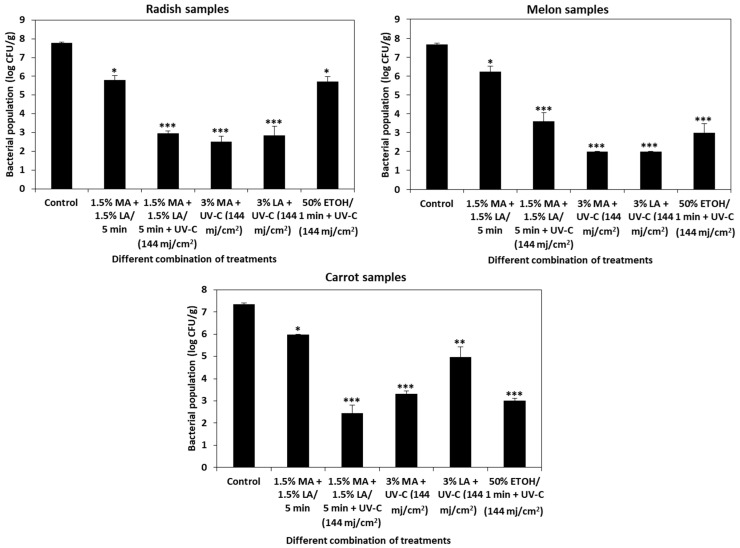
Combination of organic acids, UV, and ethanol (dipping) in cut vegetables. The effectiveness of dipping of cut vegetables in 1.5% malic acid + 1.5% lactic acid + UV-C (144 mj/cm^2^), 3% malic acid + UV-C (144 mj/cm^2^), 3% lactic acid + UV-C (144 mj/cm^2^), and 50% ethanol + UV-C (144 mj/cm^2^) was tested for its antilisterial activity in radish, melon, and carrot samples. The data obtained are the means ± SD from three independent experiments with triplicates. Statistical analysis (*t*-test) was conducted using SigmaPlot. For t-test analysis, the values of the dip-treated cut vegetable samples were compared with those of the untreated cut vegetable control samples. An asterisk indicates significant differences compared with the control (* *p* < 0.01, ** *p* < 0.001, *** *p* < 0.001).

**Figure 6 foods-10-02135-f006:**
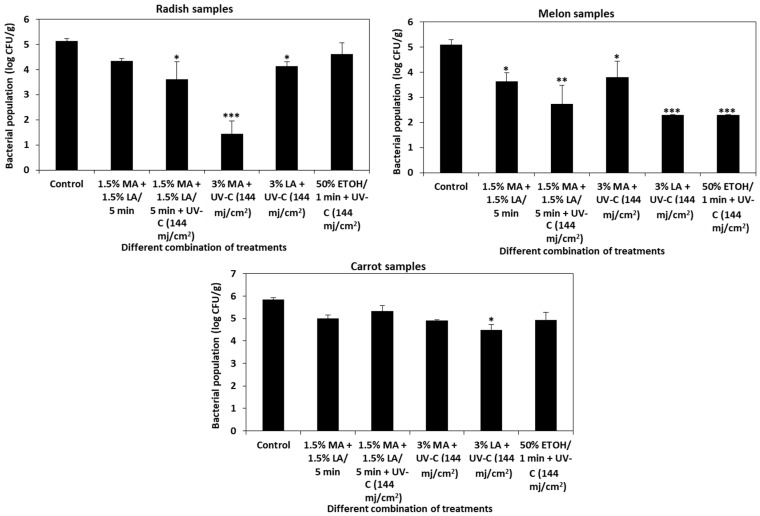
Combination of organic acids, UV-C, and ethanol (spraying) on intact vegetables. The effectiveness of spraying 1.5% malic acid + 1.5% lactic acid + UV-C (144 mj/cm^2^), 3% malic acid + UV-C (144 mj/cm^2^), 3% lactic acid + UV-C (144 mj/cm^2^), and 50% ethanol + UV-C (144 mj/cm^2^) on intact vegetables was analyzed for its antilisterial activity in radish, melon, and carrot samples. The data obtained are the means ± SD from three independent experiments with triplicates. Statistical analysis (t-test) was conducted using SigmaPlot. For t-test analysis, the values of the spray-treated intact vegetable samples were compared with those of the untreated cut vegetable control samples. An asterisk indicates significant differences compared with the control (* *p* < 0.01, ** *p* < 0.001, *** *p* < 0.001).

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
