# Peer review of "Development of Strategies to Minimize the Risk of Listeria monocytogenes Contamination in Radish, Oriental Melon, and Carrots"

_foods, 2021, doi:10.3390/foods10092135_

Round 1

Reviewer 1 Report

This manuscript describes the effect of organic acids, ultraviolet-C light, and ethanol on the reduction of Listeria monocytogenes inoculated on radish, Oriental melon and carrot samples, both whole and cut. Specifically the authors were interested in the efficacy of treatments at different concentrations, when applied individually and in combination. While this manuscript explores an important area of research, more details and clarifications are needed for data presentation, statistical analyses, and other sections. In addition, thorough proofreading is required for grammatical errors, sentence structure and overall clarity.

Overall concerns

A number of treatments and samples are described, however, inadequate detail and lack of clarity in the materials and methods, and subsequently in the presentation of results make it difficult to understand what was done and what is actually being reported. Sample treatments and some of the treatment differences are not clear, especially how combinations of treatments are applied, in which order etc. The results do not clearly explain which data correlate to the cut produce studies and which correlate to intact produce. Throughout the results section numbers are provided in brackets without clarifying what these numbers represent. Are they log CFU/ml? Are these numbers the population left after the treatment or do they represent the population reduced? The manuscript needs to be restructured and some sections need to be expanded upon for clarity. The discussion needs to be rewritten to reflect what was done. The authors conclude that the differences in the effectiveness of sanitizing treatments are “mainly due to the difference in the characteristics of the food surfaces” yet they do not provide any evidence of this, and have no data to support this statement.

Specific comments

Major issues

  • Grammar and proofreading issues, including awkward wording choices (e.g., line 10 – develop, lines 117, 119 - treated on), capitalization issues (e.g., lines 36 -37), and incomplete/confusing sentences (e.g., lines 42-44). There are issues with contradictory statements (e.g., line 62 – organic acids are chemicals so how do these differ from “chemical sanitizers”).
  • Some references are provided in the introduction but the text accompanying these references does not match (e.g., line 45 – biofilms are not mentioned anywhere in the manuscript referenced and line 70 – environmentally friendly is not mentioned with regard to UV-C treatments).

Introduction:

The introduction contains scientifically incorrect statements and therefore does not present the necessary background information and logical argument for this research. For example, in line 36, hard-boiled eggs are not considered produce items. Another example is in lines 52-60, when numerous non-thermal and chemical processes are called out as methods to control L. monocytogenes. However, if this paper is on produce, almost all references are related to L. monocytogenes control in meat and cheese products. Disinfection-by-products and chemical residues are discussed as a potential negative implication of these treatments. However, this would not apply to most of the non-thermal processes mentioned and at least one of these products breaks down into non-toxic compounds. It is not clear what research has been done on the produce items of interest in relation to thermal, non-thermal, or chemical treatments outside of this study.

Line 68- preferred method is subjective with regard to UV-C treatments.

Materials and Methods:

All sections of the methodology are unclear and need more detail. For example, why were three dimensions considered for one produce item but not the other two (lines 97-98)? Other initial questions include:

  • What was the temperature for glycerol stock storage?
  • Is it true that only one transfer from glycerol to TSBYE was done prior to Lm usage in experiments?
  • What was the pH of the PBS solution?
  • Report centrifugation conditions using xg as opposed to rpm, and provide instrument reference.
  • Tryptic (Line 90) and Phosphate (Line 93) should not be capitalized.
  • How were counts determine for the corresponding 0.5 OD value provided in line 95.
  • Unclear when Lm overnight cultures were brought together as a cocktail (before or after centrifugation).
  • How was the protocol determined or what reference was used for the 70% ethanol for five-minute step prior to treatments of produce? Based on results shown below, this may not be effective at removing pathogens of concern prior to lab inoculation.
  • Line 99- was the DW water sterilized prior to use?
  • Need to separate out details for intact versus cut produce. Was a size range considered for intact produce?
  • Why were two different inoculation methods used between the intact and cut produce items (i.e., spray versus spot; lines 102-106)?
  • How were the intact produce items peeled? How much of the peel was sampled?
  • Line 44- provide a reference for “1 - to 2-Log”
  • Line 89- serotype is missing for 3rd

It is unclear what the control was in this study. Was the only control the L. monocytogenes suspension in the centrifuge tube (lines 105-106)? More controls than this are required for a complete study design. Were non-inoculated produce items sampled as a negative control? Were all produce items for the same biological replicate collected on the same day from same vendor/area?

The statistics section is also needing more detail. As written, it is unclear as to what variables were analyzed in this study and what analysis was done prior to using post hoc testing procedures.

It is not clear why a different concentration of organic acid (i.e., 1.5% lactic or malic acid) was included in the combinatory treatments as opposed to using previously tested concentrations (Section 3.4; lines 205-209). Single application experiments only looked at 1 and 3% organic acid applications.

It is unclear why sample sizes were also changed to cubes and what the sizes of these cubes were for this set of experiments.

The application of organic acids or ethanol by a dipping method (which differs from the single treatment applications) is not explained. By changing inoculation methods, direct correlations between single and combination treatments cannot be made. Spraying and spot inoculation methods were also used for the initial experiments mentioned in the manuscript.

The temperatures at which these treatments were applied in this study was not provided, yet the important of L. monocytogenes attachment, growth, and survival being dependent on storage time and temperature is the first point in the discussion section (lines 252-254).  Please provide the temperature(s) used.

Results:

Sentences in the results are repeated from the methodology and should be removed from the results section (e.g., 144-146 and 209-210). There is no discussion of significant values from the statistical analysis provided in the results section. Some of the results provided do not match the statistics provided in the figures. For example, no differences were mentioned for carrot samples in Figure 1; however, different subscript values are present over several treatment bars in this figure (see lines 152-154 and Figure 1).

It is unclear in the results and figures when the authors are presenting results for the intact versus the cut produce items.

How are chemical sanitizers different from organic acids? (line 263)

Discussion:

Some statements in the discussion belong in other sections of the manuscript. For example, lines 256-259 and 285-287 belong in the introduction. The final paragraph of the discussion section (lines 336-344) is speculative as surface characteristics were not a major factor explored in this study.

Line 276 – superior to what? Are they superior based on the data? Only 2-3 log reduction is reported for radishes, and what about carrots?

Reviewer 2 Report

Authors describe in their paper different treatments applied to reduce L. monocytogenes population in spiked radish, oriental melon and carrots. They also combined treatments for getting a higher reduction of bacterial population. However, the treatments discussed in the paper cannot be applied as a general practice for all types of fresh produce, but being customized for each crop to prevent cross contamination during post-harvest washing.

Major comments:

Line 87-106. Where the fresh produce, after cutting and disinfection procedure, checked for initial contamination with possible Listeria monocytogenes? Given the fact that different produce employed in the paper have different water content, I am not sure that the oriental melon was completely dry before and after inoculation with the bacterial suspension. For example, in case of melon, which has a high content of water and would not dry as much as intended, wouldn’t be the case that not allowing a 24 h attachment of bacteria to adhere to the surface, after dipping in organic acids, would wash away all the bacteria? Hence, getting a significant reduction when applying different treatment involving dipping method? It could the case or not, have you considered this aspect?

Line 95. Was this checked as well by plating method?

Line 203. Perhaps a schematic figure would be more appropriate to describe the different combinations employed.

Minor comments:

Line 36. Please use small letters, not capital letters.

Line 49-50. This phrase is too general. Are you saying that we cannot rid of Listeria if we wash the produce only with water?

Line 58. Missing word “that could affect..”

Figure 4. Missing some time periods when UV treatment was applied. Why did you choose particular these combinations?

Round 2

Reviewer 2 Report

The manuscript has been considerably improved.

Author Response

Authors thank the reviewers for their valuable comments and suggestions that helped to improve the quality of this manuscript.